# Evaluation by Proton-Radiation Tests of a COTS-Embedded Computer Running the cFS Flight-Mission Software for a Nanosatellite

**DOI:** 10.3390/s25247661

**Published:** 2025-12-17

**Authors:** Vanessa Vargas, Pablo Ramos, Alfredo Bautista, Alejandro Castro-Carrera, Yolanda Morilla Garcia

**Affiliations:** 1Grupo de Investigación Embsys, Departamento de Eléctrica, Electrónica y Telecomunicaciones, Universidad de las Fuerzas Armadas—ESPE, Av. General Rumiñahui y Ambato, Sangolquí 171103, Ecuador; pframos@espe.edu.ec (P.R.); afcastro@espe.edu.ec (A.C.-C.); 2Grupo de Investigación Embsys, Carrera de Ingeniería en Electrónica y Automatización, Universidad de las Fuerzas Armadas—ESPE, Av. General Rumiñahui y Ambato, Sangolquí 171103, Ecuador; 3Centro Nacional de Aceleradores (CNA), Universidad de Sevilla, Consejo Superior de Investigaciones Científicas (CSIC), Junta de Andalucia, 41092 Sevilla, Spain; ymorilla@us.es

**Keywords:** accelerated testing, SEE, SEFI, SEU, on-board computer, embedded computer

## Abstract

This work aims to evaluate the feasibility of using a COTS-embedded computer as an on-board computer (OBC) for nanosatellites in academic projects. The prototype is based on the BeagleBone Black board, which runs the cFS flight-mission software on the RTEMS operating system. For evaluation purposes, 15.9 MeV proton-accelerated radiation tests were performed at the CNA facility to obtain the soft-error rate of the DDR3 SDRAM. Results show the presence of bit-flips in memory cells, leading to error propagation, and a burst of errors produced by SEEs, affecting the control logic of the SDRAM memory. Despite the errors and accumulated dose, the board continued to function normally, with a worst-case FIT indicating that one failure every two years is expected in the SDRAM memory. This study suggests the possibility of using BeagleBone Black as an OBC for LEO. In addition, the article provides clues on how redundancy-based fault tolerance can be implemented.

## 1. Introduction

The increasing demand for cost-effective and rapid access to space has driven a growing interest in the use of commercial off-the-shelf (COTS) embedded computers for nanosatellite missions. While these devices offer significant advantages in terms of availability, performance, and reduced development costs, their use in space environments presents significant challenges. For example, nanosatellites operating in low-Earth orbit (LEO) are constantly exposed to radiation effects, thermal cycling, and other harsh conditions where even minor faults or errors can compromise mission success. For this reason, reliability and fault tolerance are critical factors when integrating COTS hardware into space systems. In addition, the continuous evolution of space applications demands greater processing capacity and system complexity, as on-board computers are required to support advanced payload management, autonomy, and real-time decision making [1,2,3]. Therefore, the design of satellite platforms must be based on the balance between the risks of using COTS devices and the benefits they provide.

As COTS devices are intrinsically more vulnerable to the severe radiation conditions present in space compared with their radiation-hardened counterparts, a mandatory step is to evaluate the possibility of using them in LEO and CubeSat applications. Among the various radiation threats, proton-induced single-event effects (SEEs) are a predominant source of errors that can compromise both hardware integrity and software operational reliability [4,5]. Consequently, it is imperative to perform proton-radiation testing for quantifying the sensibility of COTS-embedded computers to single-event upsets (SEUs), single-event functional interrupts (SEFIs), single-event latch-ups (SELs), and total ionizing-dose damage [6,7,8].

A substantial challenge in the proton-radiation testing of complex COTS systems is the precise identification of fault origin. The classification of errors as either transient or permanent is often challenging, a problem that is exacerbated when non-recoverable failures originate from peripheral components such as flash memory [9,10]. In the case of an embedded architecture using a real-time operating system, the complex interaction between hardware and software further complicates the identification of the fundamental cause of a system malfunction, interfering with efforts to attribute fault responsibility and interpret system-error logs [11].

In this context, this article aims to evaluate the sensitivity to SEEs of the DDR3 SDRAM of a COTS-embedded computer implementing the Core Flight Software (cFS) V6.7.0 of NASA running on RTEMS, by means of proton-radiation ground testing. The main contribution of this study is to provide the soft-error rate (SER) of the embedded DDR3 SDRAM at 15 MeV proton energy. Furthermore, the article presents a comprehensive explanation of the observed errors, which provides information on how fault-tolerant strategies can be applied.

### Related Works

Authors of [6] present a structured methodology for assessing the radiation tolerance of a CubeSat on-board computer (OBC) with high-performance payloads operating in harsh orbital environments. This work details a comprehensive radiation-test campaign, beginning with a total ionizing-dose (TID) evaluation to 50 krad using a Cobalt-60 source. Then, the report details SEE studies for characterizing, in detail, the central microcontroller using a component-level pencil beam. Finally, the study synthesizes the findings from the entire campaign to propose specific mitigation strategies. A significant procedural contribution is the development and validation of a dedicated methodology for performing SEE characterization at medical accelerator facilities.

Authors of [12] present an analysis of SEEs in the NVIDIA Jetson Orin NX system on module (SoM). The main contribution of this study is a comprehensive proton-irradiation-test campaign that evaluates the susceptibility of the entire module, including its central CPU, GPU, and integrated peripherals. A significant methodological advancement has been achieved through the effective implementation of ARM’s reliability, availability, and serviceability (RAS) subsystem in the diagnosis of errors. This advancement enables precise identification of the origins of single-event functional interrupts (SEFIs) at the SoM level. This technique enables the direct observation and detailed characterization of SEEs within the GPU complex. The study concludes with a consolidated assessment of the SoM’s overall radiation susceptibility, pointing out its most sensitive components.

The article [13] offers critical experimental data on the radiation susceptibility of a commercial SRAM-based FPGA, specifically the Arria 10 model, by characterizing its response to proton-induced SEEs. A significant methodological contribution is the systematic testing conducted across three distinct proton energy levels, which facilitates an analysis of how SEE cross-sections vary with incident energy. The most significant finding is the identification of an important sensitivity to protons with energies below 50 MeV, a result with direct implications for risk assessments in low-Earth orbit, where lower-energy protons are abundant. Consequently, this work delivers essential validation data on a specific COTS component and underscores the importance of including lower-energy proton spectra in test protocols to accurately bind the error rates for FPGA-based systems in the space environment.

The research described in [7] presents a comprehensive characterization of proton-induced single-event effects in the on-chip SRAM of an ARM Cortex-A9 MPCore-embedded processor (NXP LQFP100, Natong, China). The primary contribution of this study lies in the detailed reporting of proton-beam test results, including the acquisition of energy-dependent cross-section data that quantifies the memory’s susceptibility. Beyond empirical observation, the study’s significance is elevated through the development of probabilistic fault models derived directly from the experimental data. These models, which encompass associated fault distributions, establish a foundational framework for system-level reliability analysis. The practical utility of this approach is demonstrated through its application in fault-injection campaigns on a software-benchmark suite, thereby bridging the gap between physical radiation effects and their system-level functional consequences.

Authors of [14] studied the proton-radiation effects on DDR5 SDRAM modules including SEEs and accumulated effects, considering several factors such as proton energy, module manufacturers, and the specific power-management unit. Experiments showed that the peak $\sigma_{SEE}$ and $\sigma_{SEFI}$ occurred near 25 MeV energy. Furthermore, a comparison between server-grade DDR4 and DDR5 modules was performed to analyze the impact of different generations of memories involving an external error-correction code and accumulated effects. The main objective of this research is to determine the radiation performance of DDR5 as it is a promising candidate for future space applications.

## 2. Materials and Methods

The present work focuses on the evaluation of an OBC computer implemented on a single-board embedded computer running the cFS mission software (CFE V6.7.0). The selected computer is the BeagleBone Black Rev C, running the RTEMS operating system (OS) V6.0.0. On top of the OS, the flight-mission software was configured and a user application was developed. For evaluation purposes, radiation ground tests with 15 MeV proton were performed to determine the device suitability for specific missions or space applications.

### 2.1. Embedded Computer

The low-cost BeagleBone Black (BBB) COTS board is based on the Sitara SOC AM3358BZCZ (Texas Instruments, Hsinchu, Taiwan), operating at 1 GHz [15]. It has an ARM^®^ Cortex-A8 32-Bit RISC processor. It has, as main peripheral interfaces, 4 × UART, 2 × SPI, 2 × I^2^C, 2 USB 2.0 high-speed DRDs (dual-role devices), two industrial gigabit ethernet, 2 CAN ports, 32 GPIO per bank, 3 SDIO ports, 32 timers, and ADC [16]. This flexibility makes it suitable for different applications. The board is provided with a 512 MB DDR3 SDRAM memory, which is one of the most sensitive areas for SEEs. Figure 1 depicts the BeagleBone Black board, and Figure 2 shows the BeagleBone Black architecture, with Sitara SoC being the main component. The present study is devoted to analyze how radiation effects affect the performance of a memory-bound application implemented over the cFS software.

#### Sitara SOC

The Sitara SoC includes various types of memories: a 32 kB of L1 instruction and 32 kB of data cache with single-error detection (parity), 256 kB of L2 cache with error correcting code (ECC), 176 kB of on-chip boot ROM, 64 kB of dedicated RAM, and 64 kB of shared L3 RAM. In addition, the chip includes an external memory interface (EMIF) to manage up to 1 GB of external SDRAM [17]. Figure 3 schematizes the architecture of the Sitara SoC. It can be observed that the memory interface controls the external DDR3 through the mDDR subsytem. The block diagram of the mDDR is illustrated in Figure 4. This subsystem manages read and write operations from or to the external DDR.

As described in the manual, within the SoC memory map, the SDRAM is located at the address 0x80000000. The BBB version C used in this work, implements an external SDRAM Kingston D2516ECMDXG of 512 MB, configured as 32 M ×16 bits ×8 banks. Thus, 256 M words occupy the address ranges from 0x80000000 to 0x8FFFFFFF.

### 2.2. Software Configuration

In this section, the software configuration is detailed. Concerning the on-board software, the system considers a stack comprising the following:Real-Time Operating System (RTOS)Mission-Control System (MCS)User applications

The selected RTOS was RTEMS (real-time executive for multiprocessor systems), because it was identified as the most suitable option considering cross-platform compatibility, minimal memory footprint, deterministic real-time performance, availability of comprehensive technical support, and open-source accessibility. In addition, RTEMS offered extensive documentation, demonstrated reliability, and had a proven record of deployment in critical aerospace missions conducted by the European Space Agency (ESA). This legacy of successful utilization highlights its robustness and validates its suitability for time-sensitive, resource-constrained, and safety-critical applications in the space domain.

Regarding the MCS, the core flight system (cFS) was selected due to its compatibility with the standards established by the Consultative Committee for Space Data Systems (CCSDS), ensuring interoperability and compliance with internationally recognized protocols. Furthermore, the cFS provides a high degree of portability across multiple operating systems, including Linux, RTEMS, and VxWorks, which facilitates its integration into diverse mission environments. Its low resource consumption represents a significant advantage in embedded and spaceborne systems, where computational efficiency is a critical constraint. In addition, the cFS has become a widely adopted framework within aerospace applications, reinforcing its reliability and maturity. Its distribution under an open-source license, and active maintenance by NASA, ensure sustainability, continuous support, accessibility, and opportunities for community-driven development. Therefore, the software proposal is shown in Figure 5.

RTEMS was installed on the BBB platform using the RTEMS source builder (RSB), which was a tool designed to generate the essential RTOS components along with the board support package (BSP). The BSP provides the support layer that interfaces the operating-system kernel with the processor hardware, ensuring the proper initialization and integration of the underlying architecture.

To meet the communication requirements of the core flight-system (cFS) framework, the LibBSD library was integrated into RTEMS, enabling support for network protocols, such as UDP/IP, which are essential for ensuring telemetry transmission and command reception from the ground station. On the top of this infrastructure, NASA’s cFS was deployed, providing a modular environment for the execution of user-defined applications for several purposes.

### 2.3. User Application

One of the most sensitive areas of the OBC to space radiation (SEEs) is the SDRAM memory. In order to evaluate its sensitivity, matrix multiplication was proposed as a memory-bound application.

For this purpose, two matrices, A and B, were defined, each one with dimensions of 1670 × 1670. Each matrix element is an unsigned integer of 32 bits. Matrix A elements were initialized with a constant value of 0x00000001, while B’s elements were initialized with 0x00000002. The matrix multiplication process was executed 11 times to fill the memory with the resulting matrices, C0 to C10. After operation, each Cn’s element should have a value of 0x00000D0C.

The dimensions of the matrices were chosen considering that the main objective was to occupy the maximum space of memory to check bit-flips during radiation tests. The flow diagram of the application is illustrated in Figure 6. In order to facilitate the analysis, testpoints were added in the application’s flow diagram.

After the execution of 11 replicas of matrix multiplication, the application verifies the integrity of data stored in the memory. First, it checks if matrices A and B retain their initialization values. Then, the application verifies the resultant matrices C0 to C10, confirming the expected values. If inconsistencies are identified, the program logs the memory address and the read value where the error occurred. Finally, the program runs again, starting with matrix multiplication and then the verification process.

### 2.4. Adopted Test Approach

To perform a realistic characterization of the space harsh environment, proton-radiation ground tests were carried out. The methodology is as follows:Select the proton facility to run the testsEstablish the proton fluencyImplement the experiment setupDefine the test procedureRun the testsAnalyze data

#### 2.4.1. Proton-Radiation Facility

These tests were run at the Centro Nacional de Aceleradores (CNA) facility in Seville, Spain, using the 18/9 cyclotron [18]. This accelerator supplies protons with a fixed nominal energy of 18 MeV. However, the external irradiation beam line associated with the cyclotron was used for the experimental campaign. Figure 7 shows a schematic view corresponding to the external line of the cyclotron at the CNA.

#### 2.4.2. Experiment Setup

During irradiation experiments, physical access to the radiation chamber is prohibited. Therefore, a support system was implemented to monitor and control the experiment. First, between the DUT and the power supply unit, an anti-latch-up circuit was placed to protect the DUT from currents exceeding a predefined safety threshold. A microcontroller board controls the device turning on and off and the reset. When a single-event latch-up (SEL) is detected, the system automatically interrupts power delivery to the DUT. This board was controlled via UART serial communication from a host computer located outside of the radiation chamber. In order to monitor and log experimental data, the DUT was connected to the host computer through another serial connection.

The DUT was positioned at a distance of 490 mm from the beam exit nozzle. To precisely define the irradiated area and reduce the fluence of the beam halo on unwanted regions of the sample, a 100 μm thick grounded aluminum mask was placed. It had a 25 mm diameter circular hole that served as an exit window between the beam and the DUT. The distance between the DUT and the beam allowed for the intervening air layer to act as a natural energy degrader. Under these conditions, the effective energy of the protons incidentally on the DUT surface was reduced to 15.9 MeV, with an estimated energy dispersion of approximately 500 KeV.

The incident proton flux on the device under test (DUT) was monitored indirectly since no continuous current readout was available at the DUT level. For this purpose, during irradiation, the beam current was measured through an electrically isolated graphite collimator positioned immediately downstream of the cyclotron exit window. Data acquisition was performed in real time using a Brookhaven Model 1000c current integrator (Brookhaven Instruments Corporation, New York, NY, USA) set to the high-sensitivity range (0.05 nA within a 2 nA range) to ensure adequate resolution for the low currents characteristic of irradiation tests. In order to accurately determine the effective proton flux at the DUT surface, a prior calibration was required, incorporating correction factors associated with the emission and backscattering of secondary electrons. The setup of the test experiment is illustrated in Figure 8.

To reduce variable factors affecting the experiment, the temperature and humidity were kept constant with values of 22 °C and 40% RH in the chamber. The flux range considered for the experiments was between $2.6\times 10^{7}$ p · ${\mathrm{cm}}^{-2}\cdot$ s^−1^ and $3.1\times 10^{7}$$n\cdot {\mathrm{cm}}^{-2}\cdot s^{-1}$. Sections 3 and 6 of the JESD89B document of the JEDEC STANDARD will be used as the base protocol for the tests [19].

#### 2.4.3. Device Under Test

The target device was the DDR3 DRAM memory (Kingston D2516ECMDXGJD) of the BeagleBone Black rev C. This is a 4 Gb CMOS dynamic random-access memory configured in 32 M words × 16 bits × 8 banks. Its technology process is 78 nm, requires a VDD of 1.35 V, and supports a data rate of 1866 Mbps (double-data-rate architecture, two data transfers per clock cycle). In addition, it has a multi-purpose register (MPR) for predefined-pattern read out to allow link and data training.

This DRAM memory has 2 kB page size, with 15 bits for row address (A0 to A14), 10 bits for column address (A9 to A0), and 3 bits for bank address (BA0 to BA2). Each bank is organized in 32 K rows, and each row stores 128 blocks. Each block contains the information of 8 words (8×16 bits =128 bits). For addressing a block, column address A3 to A9 are used.

During reading operation, the data block is stored in the READ FIFO. Then, by using column address A0 to A2, data of the selected word is sent to the READ DRIVERS. In the writing process, data bus information is loaded into the WRITE driver. Then, 16 bits are sent to the data interface. Once the data interface has the information of a complete block, it executes the writing process into the respective bank. Figure 9 depicts the functional block diagram of the DDR3L memory.

#### 2.4.4. Test Procedure

In this study, two tests were defined. A first test of 140 min was planned with a flux of $2.7\times 10^{7}\;p\cdot s^{-1}\cdot {\mathrm{cm}}^{-2}$; then, there was a break time of half the irradiation time (70 min), and after that, a second test of 140 min with a flux of $3.0\times 10^{7}\;p\cdot s^{-1}\cdot {\mathrm{cm}}^{-2}$. The break was proposed to let the DUT recover. During the tests, the DUT was in active mode. Each test began with the initialization of the board with the proton beam turned off. Once the program was running, it began the irradiation. During the tests, the Sitara SoC controlling the memory logged data to the host computer. It saved information in database files for further analysis. Data logged included the time stamp, testpoints in the application, and errors.

As was mentioned, the selected application was a matrix multiplication with multiple spatial and time redundancies. The same input matrices were operated 11 times and resulting matrices were saved in different memory banks of the memory. Figure 10 depicts the location of matrix data in SDRAM. It is important to note that matrices C1 to C10 start from a predefined physical memory address, while matrices A and B and C0 are dynamically addressed by the operating system.

By analyzing the resulting data, it is possible to determine the application soft-error rate. In addition, the results allowed the evaluation of how many copies were necessary to improve fault tolerance in this type of COT device, and in which memory regions the copies had to be placed to reduce consequences.

## 3. Results

In this section, the results of two radiation tests performed on the DUT are presented.

### 3.1. First Test

In the first test, a flux with a mean of $2.7\times 10^{7}\;p\cdot s^{-1}\cdot {\mathrm{cm}}^{-2}$ was applied during 7949 s, resulting in a fluence of ∼$2.1\times 10^{11}\;p\cdot {\mathrm{cm}}^{-2}$. During this test, six errors occurred. The first and third were bit-flips, the second one was a SEFI, and the others were bursts of errors. The detailed error analysis is explained below.

#### 3.1.1. Bit-Flips

During this test, two single-bit upsets (SBUs) occurred. In both cases, the affected bit changed from ‘0’ to ‘1’. From log analysis, it could be established that the first bit-flip occurred after testpoint *t10* while the second occurred after testpoint *t4* of flow diagram depicted in Figure 6. Table 1 summarizes the SBU information.

#### 3.1.2. SEFI

During a second run of the application, a SEFI occurred at exposure time 00:16:52.7, when it was executing the matrix multiplication of C0. Therefore, it was necessary to reboot the system. This was performed through the host application.

#### 3.1.3. Bursts of Errors

Three burst of errors were logged in the first test. The first burst of errors starts at matrix element A[1551][1614] with processor address 0x85802000 and ends at matrix element B[1135][492] with address 0x85ffe008. Each matrix element is 32 bit-data, using four memory locations for storing information. A sample of the first burst of errors is summarized in Table 2.

In Table 2, it can be seen that logged errors follow a pattern with regard to memory addresses and data values. The analysis of the addresses that are involved in the first burst of errors is summarized in Table 3. For better illustration, cyan and blue color bits represent column addresses (A0 to A9); green color bits represent the address bank (BA0 to BA2); and red and brown represent the row addresses (A0 to A14). The pattern comprises fixed and changing-bit address lines. The fixed are the brown and green, with row addresses A[14:10] equal to 0b01011; A0 equals ’1’, and bank addresses equal zero. The changing bits are the red color bits, sweeping all the possible combinations of lines A1 to A9 (from 0b 000 0000 0000 to 0b 111 1111 1111), combined with four values of column addresses: 0b00 0000 0001, 0b00 0000 0011, 0b00 0000 1001, and 0b00 0000 1011. Therefore, the first burst logs 1024 matrix elements with erroneous values in two bytes, giving a total of 2048 errors in the data read with values of 0xff.

It is important to note that the logged errors were located in matrices A or B; therefore, an error propagation would be expected in the resulting matrices. However, it did not occur. Thus, it is presumed that the burst error was provoked by an error during the reading process, probably caused by a SEU on a reading control register.

A second burst of error appeared when reading the C1 [1561][27] element located at address 0x87801cc0 and ending at address 0x87991ccc. In this case, the burst started at C1 of the resulting matrix and ended at C2 [136][460]. Similarly to the first burst, a pattern in the logged addresses can be established. In this case, for six sets of addresses, there were possible sweeping combinations of row address lines from A1 to A9. The log process starts at combination 0b 000000000 of the row address value (A1 to A9), but it was interrupted at combination 0b 001100100. At this point, another type of burst of errors appeared in the log. Due to this interruption, the log of the second burst consisted of 606 erroneous data with read values of 0xff or 0x00. A complete analysis is detailed in Appendix A.

The third burst error appeared when reading the C2 [138][480] element located at address 0x8799514c. In this case, the burst appeared in C2 and following the resulting matrices. The pattern is different from the two previous bursts, both in memory address and data. Regarding the memory address, this burst also had fixed and changing bits. However, the possibilities of column addresses involved were 96 and 64 times more than the first and second burst, respectively. Errors located in the element memory address ended in 0x4 and 0xC. When ending in 0x4, the erroneous data read was 0xfe020b0c, while when ending in 0xC, the erroneous data read was 0xff010c0c. Also, the complete analysis of this burst is presented in Appendix A.

Following a similar analysis of the first two bursts, it can be observed that row addresses A1 to A9 are also involved but do not begin from 0b 000000000 as the other bursts. Instead, they began from 0b 001100101, which was the next combination of the last log error of the second burst. This behavior suggested that the second burst error continued affecting the row address. In addition, the new SEE affected the bank and column addresses. This behavior is analyzed in detail in Appendix A. Furthermore, the data pattern of this burst did not have fixed values such as 0xff or 0x00. Consequently, it is assumed that this third burst was caused by a SEE affecting other hardware.

### 3.2. Second Test

In the second test, a flux with a mean of $3.0\times 10^{7}\;p\cdot s^{-1}\cdot {\mathrm{cm}}^{-2}$ was applied during 2600 s, resulting in a fluence of ∼$7.8\times 10^{10}\;p\cdot {\mathrm{cm}}^{-2}$. During this test, five errors occurred. The first four were bit-flips and the last one was a burst of errors.

#### 3.2.1. Bit-Flips

During the second test, four single-bit upsets (SBUs) occurred. In all cases, the affected bit changed from ‘0’ to ‘1’. From log analysis, it could be established that the first bit-flip occurred after testpoint *t5*, the second one after testpoint *t6*, the third after testpoint *t11*, and the last one after testpoint *t12* of the flow diagram depicted in Figure 6. Table 4 summarizes the SBU information.

#### 3.2.2. Burst of Errors

In the second test, one burst of errors occurred. Its behavior is similar to the second burst of errors. This burst of errors starts at matrix element C0 [723][1050], with processor address 0x868022b0, and ends at matrix element C1 [303][22], with address 0x86ffe2bc. This burst logs 2040 matrix elements with erroneous values in two bytes, giving a total of 3060 errors in the data read with values of 0xff or 0x00.

By analyzing the data, a pattern can be established in the logged addresses of the burst. For four fixed sets of addresses, there were all possible sweeping combinations of row address lines from A1 to A9 (from combination 0b 000000000 to combination 0b 111111111). The detailed analysis of the pattern is explained in Appendix A.

### 3.3. Application Cross Section

The results of the irradiation tests are summarized in Table 5.

From the results, the application cross-section was obtained. The DDR3L DRAM memory was placed facing the center of the target, perpendicular to the beam axis at a distance of ∼40 cm. The proton-beam energy was 15.9 MeV, and the total fluence was ∼$2.98\times 10^{11}\;p\cdot {\mathrm{cm}}^{-2}$. The cross section is defined as(1)$$\begin{array}{c} \sigma =\frac{Number\;of\;Upsets}{Fluence} \end{array}$$

A total of 10 SEUs and one SEFI was detected within the two tests. SEUs that provoked error propagation occurred in bank 0 and bank 7 of the DDR3 memory. It is assumed that SEUs were also produced by particles hitting the controlling registers, which is why three bursts of errors were observed.

The obtained application cross-section is(2)$$\begin{array}{c} \sigma_{SEE}=\frac{11}{2.98\times 10^{11}}=3.69\times 10^{-11}\frac{{\mathrm{cm}}^{2}}{device} \end{array}$$

Due to the scarcity of experimental data, it is compulsory to add uncertainty margins to these results. For a 95% confidence interval (α=0.05), the lower and upper limits for the SEE cross section are(3)$$1.84\times 10^{-11}\frac{{\mathrm{cm}}^{2}}{device}<\sigma_{SEE}<6.60\times 10^{-11}\frac{{\mathrm{cm}}^{2}}{device}$$

The performed accelerated tests are intended to determine the soft-error rate (SER) of the system while the device is irradiated in a proton beam of known flux. The results of this accelerated test can be used to estimate the terrestrial-cosmic-ray-induced SER for a given terrestrial-cosmic-ray radiation environment [19].

First of all, the failure rate (λ), which was the number of failures per unit of time, was evaluated. For the present study, (λ) can be estimated by extrapolating the obtained application cross section at LEO-inclined polar orbit with an altitude of 340 Km, where the proton flux (ϕ) is ∼229 p $\cdot s^{-1}\cdot {\mathrm{cm}}^{-2}$. It is achieved by applying the following equation:(4)λ=σ∗ϕ
$$\lambda =\left[1.84-6.60\right]\times 10^{-11}\frac{{\mathrm{cm}}^{2}}{device}*229\;p\cdot s^{-1}\cdot {\mathrm{cm}}^{-2}\;\lambda =\left[0.42-1.51\right]\times 10^{-8}\frac{p}{s}=\left[1.51-5.44\right]\times 10^{-5}\frac{p}{h}$$

Once the failure rate is obtained, the SER can be expressed as failure in time (FIT). FIT is a specific rate for electronic components, representing failures per billion ($\times 10^{9}$) hours of operation.(5)$$FIT=\lambda *1\times 10^{9}$$FIT=[15,120−54,400] failures per billion hours of operation.

The worst-case FIT value indicates that the flight system implemented on the BeagleBone Black is expected to have around 0.5 failures per year in the DDR3 SDRAM. This result does not include any fault-tolerance strategy to mitigate errors. Therefore, when applying temporal redundancy by replicating data in different banks of the memory, errors provoked by SEUs affecting the memory will disappear. In contrast, errors produced by SEUs affecting the control register or other sensitive areas of the board and SEFI’s will remain. Considering that the average lifetime of a CubeSat in LEO is about 2–5 years, it is feasible to use BBB as the on-board computer of a CubeSat for academic projects.

## 4. Discussion

During proton-radiation tests, six bit-flips in the SDRAM memory that cause error propagation have been clearly identified. In addition, four bursts of errors were observed during the tests. The first two have similar behaviors, taking into account addresses and data with erroneous values of 0x00 and 0xff. Since the SDRAM architecture has control logic to fix the read data output to a predefined pattern of 0x00 or 0xff, it is possible that a SEE directly affected the mode-register value in the electronics of the control logic of the memory (see Figure 9).

By analyzing the third error burst, it can be seen that this one is the result of the consequences of two SEEs affecting different parts of the hardware controlling the SDRAM. On the one hand, the SEE that produces the second burst continues affecting the result. On the other hand, the other SEE had probably affected the Sitara SoC mDDR subsystem, which controls the behavior of the SDRAM (see Figure 4). It can been explained that since the DUT is located near the Sitara SoC processor, it could thus be directly affected by the proton beam.

In spite of the bit-flips and burst of errors, the BeagleBone Black continued to operate normally. While it is true that SEFIs did not occur during any of the experiments, the probability of them occurring is very low, as the worst-case SER value indicates three failures per year. Moreover, each burst affected only one of the eight memory banks. This is an encouraging result that confirms the possibility of using this board for non-critical applications by applying fault-tolerant algorithms with replicated copies located in different banks.

In Table 6, a comparison between our work and similar ones is presented. In work [14], multi-energy tests were performed on the DDR5 SDRAM of different manufacturers. The study shows peak values in the $\sigma_{SEE}$ at energies of 25 MeV and 20 MeV, while no significant values were observed in the 45 to 100 MeV range. These results are quite close to the $\sigma_{SEE}$ obtained in our study. Hence, one relevant contribution of our work is to provide experimental results for DDR3 SDRAM sensitivity at 15 MeV energy.

In work [12], components such as SoC, CPU, and DDR5 SDRAM were exposed to 480 MeV of proton energy. Considering the resulting $\sigma_{SEE}$ for each component, the authors conclude that NVIDIA Jetson Orin NX is an excellent candidate for high-performance computation in space applications. Comparing this study with ours, it can be observed that at 480 MeV of proton energy, the occurrence of SEEs increases by a factor of 8, which is not significant given the higher capacity of the DDR5 memory. Thus, this comparison suggests that the DDR3 SDRAM could also be adequate for space applications. Authors of [6] conducted proton-radiation tests at 250 and 200 MeV in a CLIMB CubeSat OBC, focusing on the Cortex M3 processor and the whole OBC. The authors state that the OBC maintained nominal operation during the TID (50 krad) and SEE tests, and no hard errors were detected. Therefore, the OBC is expected to be fully functional throughout the CLIMB mission. Comparing these results with our experiment, our TID was about 83 krad with no hard errors, and just one SEFI.Thus, our BBB computer is expected to have good performance with regards to the accumulated dose. Furthermore, in work presented in [20], three BBB boards were tested under the Cobalto-60 source at the Kirtland Air Force Base at a dose rate of 53 radians per second. On the one hand, the results evidenced that BBB could operate without shielding until 17 krad. On the other hand, a study of total ionizing dose (TID) for LEO has determined that the trapped-electron doses in one year is around 100 krad, and it can be reduced to 1 krad per year via aluminum shielding [21]. Consequently, these studies support the idea that by shielding the BBB, it is possible to use it in LEO applications. Finally, in [13], the sensitivity of a SRAM-based FPGA is studied. There are contrasting results comparing experiments at 200 MeV. On the one hand, in [13], the $\sigma_{SEE}$ exhibits the highest value. On the other hand, in [6], no errors were detected at 200 MeV. This implies that higher energy does not necessarily mean more errors.

One limitation of the present work is the lack of multi-energy tests, since LEO proton energy ranges widely. However, in similar works presented in Table 6, it can be seen that the SEE cross section does not change significantly with regard to the SDRAM memories. While the experiments with 15.9 MeV do not allow for a final conclusion, they provide valuable evidence regarding the feasibility of using the DDR3 SDRAM in LEO applications.

## 5. Conclusions

The results obtained from proto-radiation experiments on the DDR3 SDRAM of the BeagleBone Black implementing the cFS flight-mission software suggest the possibility of using it for low-budget space missions. However, further experiments on the Sitara processor SoC and peripherals are required to have conclusive results. The obtained worst-case soft-error rate demonstrates that the DDR3 SDRAM has an intrinsic low sensitivity to SEEs, which can be improved by implementing temporal redundancy by executing the same application several times and comparing the results. In addition, to reduce SEE consequences, it is possible to implement spatial redundancy as a fault-tolerance strategy replicating data in different banks. In future works, multi-energy proton-radiation tests will be performed on the Sitara SoC and peripherals of BBB to confirm the feasibility of using this embedded computer as an OBC for LEO nanosatellite applications.

## Figures and Tables

**Figure 1 sensors-25-07661-f001:**
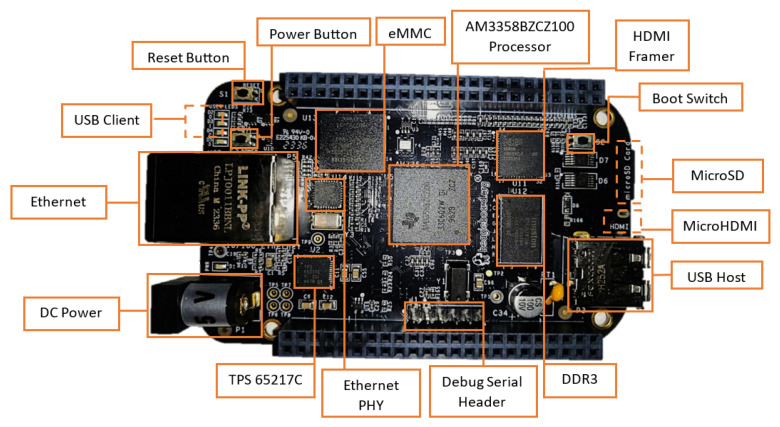
BeagleBone Black board [15].

**Figure 2 sensors-25-07661-f002:**
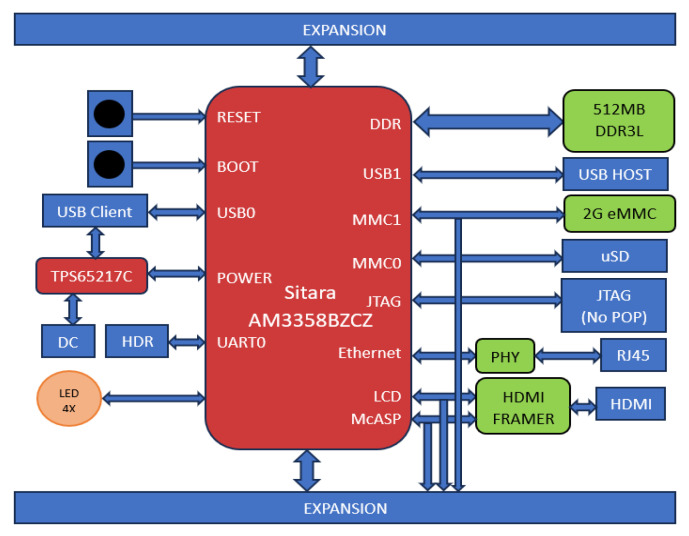
BeagleBone Black architecture [15].

**Figure 3 sensors-25-07661-f003:**
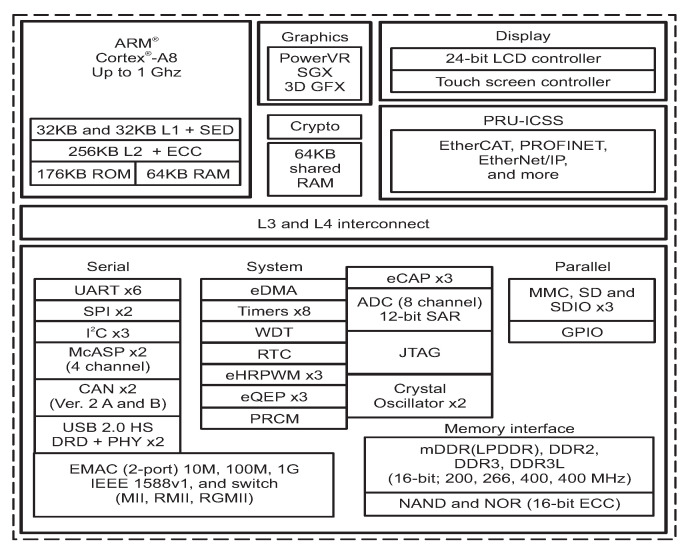
Sitara AMX335xx architecture [16].

**Figure 4 sensors-25-07661-f004:**
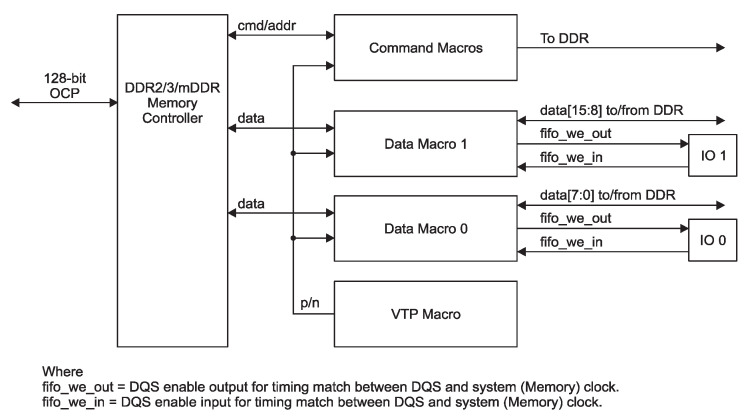
mDDR subsystem block diagram.

**Figure 5 sensors-25-07661-f005:**
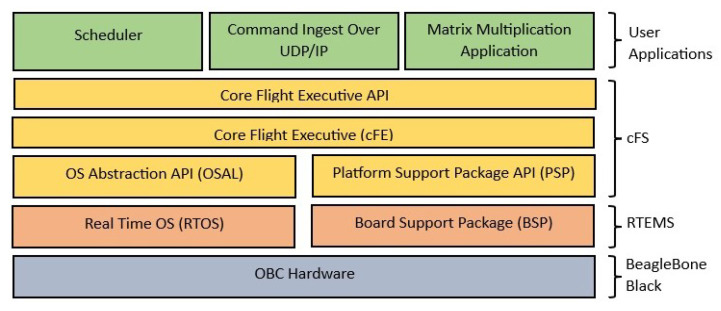
Software architecture.

**Figure 6 sensors-25-07661-f006:**
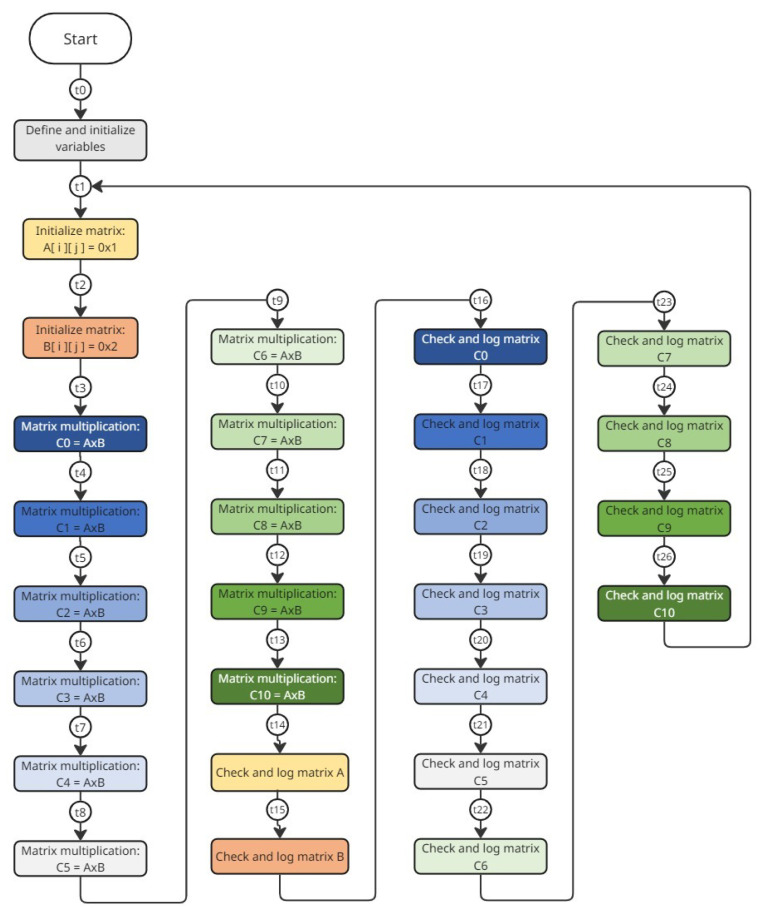
Flow diagram of user application.

**Figure 7 sensors-25-07661-f007:**
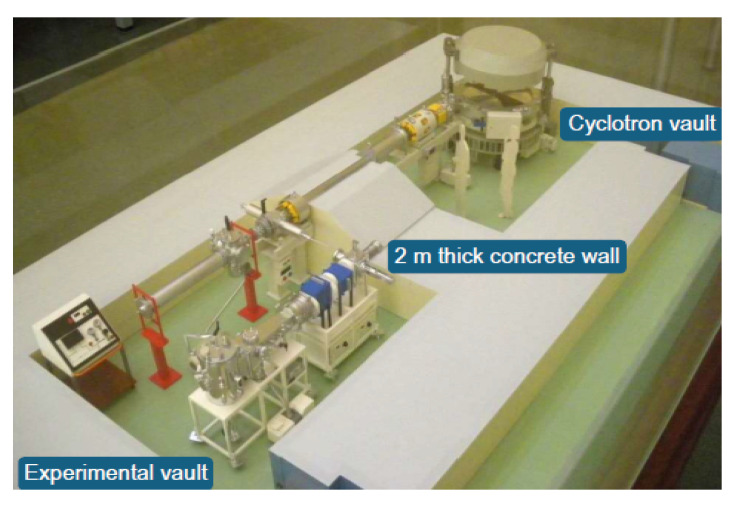
Cyclotron laboratory model.

**Figure 8 sensors-25-07661-f008:**
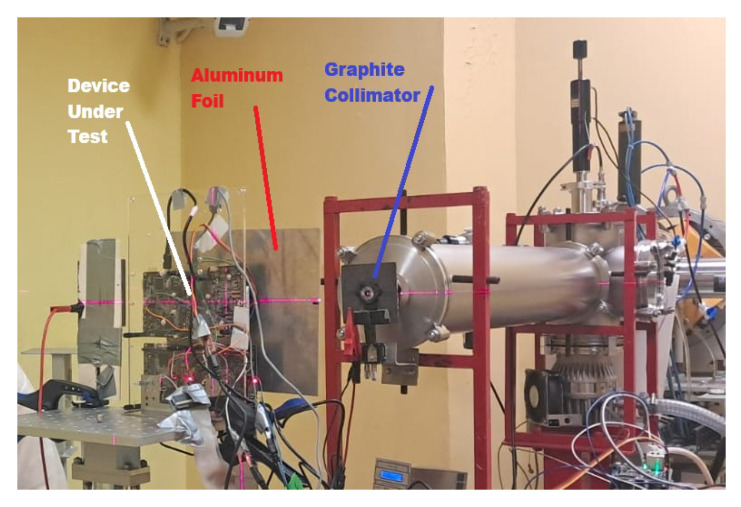
Experimental test.

**Figure 9 sensors-25-07661-f009:**
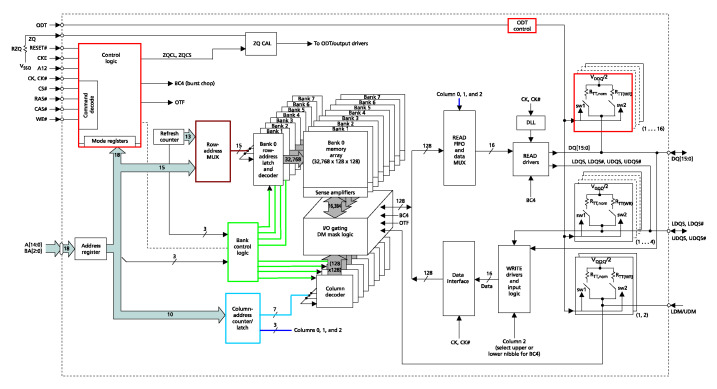
Functional block diagram of the SDRAM memory [16].

**Figure 10 sensors-25-07661-f010:**
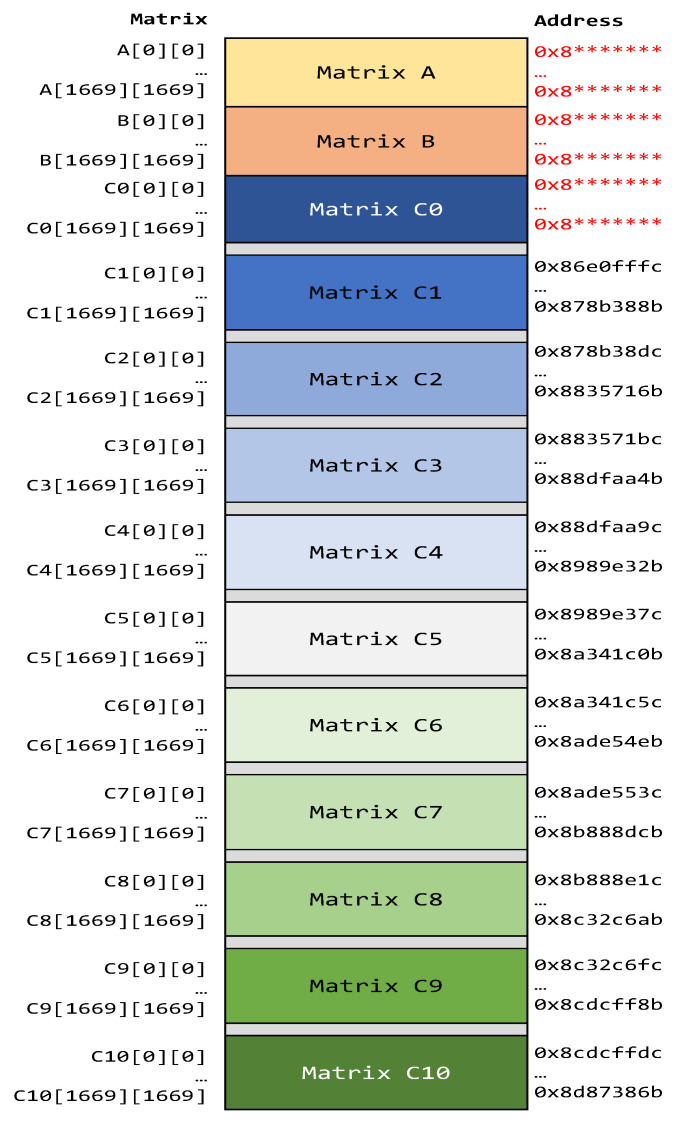
Application memory map in DRAM. Matrices A and B and C0 in red color indicate that these matrices don’t have fixed addresses. Also * symbol represents variable value.

**Table 1 sensors-25-07661-t001:** Bit-flips in first test.

Ord	Matrix Element	Located Address	Expected Value	Read Value	SBUin Bit	Resulting Matrices Element Affected		Total Propagated
						Index	From	Errors
1	B[944][409]	0x85ec66d4	0x00000002	0x00000202	9th	[i][409]	C7[16][409]	6664
2	B[1306][53]	0x86114734	0x00000002	0x0000000a	4th	[i][53]	C1[126][53]	16,576

Red color represents the erroneous read value.

**Table 2 sensors-25-07661-t002:** Burst error data in origin matrices.

Ord	Matrix Element	Element Memory Address	Memory Address	Correct Data	ReadData
1	A[1551][1614]	0x85802000	0x85802000	0x01	0x01
	*BEGIN*		0x85802001	0x00	0xff
			0x85802002	0x00	0x00
			0x85802003	0x00	0xff
2	A[1551][1616]	0x85802008	0x85802008	0x01	0x01
			0x85802009	0x00	0xff
			0x8580200a	0x00	0x00
			0x8580200b	0x00	0xff
3	A[1554][700]	0x85806000	0x85806000	0x01	0x01
			0x85806001	0x00	0xff
			0x85806002	0x00	0x00
			0x85806003	0x00	0xff
4	A[1554][702]	0x85806008	0x85806008	0x01	0x01
			0x85806009	0x00	0xff
			0x8580600a	0x00	0x00
			0x8580600b	0x00	0xff
…	…	…	…	…	…
1023	B[1135][490]	0x85ffe000	0x85ffe000	0x02	0x02
			0x85ffe001	0x00	0xff
			0x85ffe002	0x00	0x00
			0x85ffe003	0x00	0xff
1024	B[1135][492]	0x85ffe008	0x85ffe008	0x02	0x02
	*END*		0x85ffe009	0x00	0xff
			0x85ffe00a	0x00	0x00
			0x85ffe00b	0x00	0xff

Red color text represent erroneous data.

**Table 3 sensors-25-07661-t003:** Burst-error address memory.

Ord	Matrix Element	Erroneous Address	Memory Address in Binary Format			
			Offset	Row A[14:0]	Bank BA[2:0]	Column A[9:0]
1	A[1551][1614]	0x85802001	0b 1000	0101 1 000 0000 00 1	0 00	00 0000 0 001
		0x85802003	0b 1000	0101 1 000 0000 00 1	0 00	00 0000 0 011
2	A[1551][1616]	0x85802009	0b 1000	0101 1 000 0000 00 1	0 00	00 0000 1 001
		0x8580200b	0b 1000	0101 1 000 0000 00 1	0 00	00 0000 1 011
3	A[1554][700]	0x85806001	0b 1000	0101 1 000 0000 01 1	0 00	00 0000 0 001
		0x85806003	0b 1000	0101 1 000 0000 01 1	0 00	00 0000 0 011
4	A[1554][702]	0x85806009	0b 1000	0101 1 000 0000 01 1	0 00	00 0000 1 001
		0x8580600b	0b 1000	0101 1 000 0000 01 1	0 00	00 0000 1 011
…	…	…	…	…	…	…
1023	B[1135][490]	0x85ffe001	0b 1000	0101 1 111 1111 11 1	0 00	00 0000 0 001
		0x85ffe003	0b 1000	0101 1 111 1111 11 1	0 00	00 0000 0 011
1024	B[1135][492]	0x85ffe009	0b 1000	0101 1 111 1111 11 1	0 00	00 0000 1 001
		0x85ffe00b	0b 1000	0101 1 111 1111 11 1	0 00	00 0000 1 011

Colors of addresses bits are related to the memory architecture depicted in Figure 9.

**Table 4 sensors-25-07661-t004:** Bit-flips in second test.

Ord	Matrix Element	Located Address	Expected Value	Read Value	SBUin Bit	Resulting Matrices Element Affected		Total Propagated
						Index	From	Errors
1	B[769][1257]	0x85da9dac	0x00000002	0x08000002	27th	[i][1257]	C2[1530][1257]	13,500
2	B[302][1471]	0x85ab073c	0x00000002	0x40000002	30th	[i][1471]	C3[1064][1471]	12,296
3	A[879][238]	0x853b8b80	0x00000001	0x00000011	4th	[879][j]	C8[879][0]	4131
4	B[261][319]	0x85a6c764	0x00000002	0x00080002	19th	[i][319]	C9[501][319]	2839

Red color text represent erroneous data.

**Table 5 sensors-25-07661-t005:** Summary of single-event upset consequences.

Ord	Test	Type	Location	Time	Observation
1	1	SBU	B[944][409]	00:16:52.7	Occured at this time
2		SEFI	Non available	00:42:46.1	Occured at this time
3		SBU	B[1306][53]	01:20:23.2	Occured at this time
4		1st Burst	Bank 0	02:11:30.6	Logged at this time
5		2nd Burst	Bank 7	02:11:41.5	Logged at this time
6		3rd Burst	Bank 4	02:11:44.2	Logged at this time
7	2	SBU	B[769][1257]	00:01:03.8	Occurred at this time
8		SBU	B[302][1471]	00:04:46.2	Occurred at this time
9		SBU	A[879][238]	00:29:53.9	Occurred at this time
10		SBU	B[261][319]	00:33:18.9	Occurred at this time
11		4th Burst	Bank 0	00:42:36.9	Logged at this time

**Table 6 sensors-25-07661-t006:** Comparison with other works.

Reference	Platform	Device Under Test	Energy [MeV]	Fluence $\left(p\cdot {\mathrm{cm}}^{-2}\right)$	$\sigma_{\mathrm{SEE}}$ (cm^2^) Worst Case	Observations
Rodriguez et al. [12]	Nvidia Jetson GPU Orion	SoC		$\;4.41\times 10^{9}$	$\;4.5\times 10^{-9}$	First time GPU SEEs observed
		CPU	480	$\;1.99\times 10^{10}$	$\;4.11\times 10^{-9}$	
		DDR5 RAM (16 GB)		$\;1.99\times 10^{10}$	$\;5.1\times 10^{-10}$	
Koga et al. [13]	Intel Arria 10 GX	SRAM-based FPGA	50		$\;1\times 10^{-8}$	According to the authors,
		200K F-F	100	$\;2\times 10^{8}$	$\;1.5\times 10^{-8}$	FPGA is sensitive
			200		$\;1\times 10^{-7}$	to protons below 50 MeV
Treberspurg et al. [6]	CLIMB CubeSat OBC	μC Cortex M3 with	250	$\;2.45\times 10^{11}$	$\;4.8\times 10^{-11}$	
		512 kB flash and 64 kB RAM	200		0	No events observed at 200 MeV
		OBC	250	$\;3.93\times 10^{10}$	$\;2.52\times 10^{-9}$	
Li et al. [14]	Motherboard ASUS Z790	DDR5 SDRAM Kingston	20		$\;3\times 10^{-11}$	
		KVR48U40BSB-16	25		$\;5\times 10^{-11}$	$\sigma_{SEE}$ exhibit peak at 25 MeV
			45	$\;1\times 10^{11}$	$\;1\times 10^{-11}$	with a secondary peak at 20 MeV
			60		$\;2\times 10^{-11}$	
			75		$\;2.4\times 10^{-11}$	
Our Work	BeagleBone Black SBC	DDR3 SDRAM 512 MB Kingston D2516ECMDXGJD				
			15.9	$\;2.98\times 10^{11}$	$\;6.6\times 10^{-11}$	

## Data Availability

The raw data supporting the conclusions of this article will be made available by the authors on request.

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
