# Peer review of "Evaluation by Proton-Radiation Tests of a COTS-Embedded Computer Running the cFS Flight-Mission Software for a Nanosatellite"

_sensors, 2025, doi:10.3390/s25247661_

Round 1

Reviewer 1 Report

Comments and Suggestions for Authors

This manuscript addresses a practical and critical gap in low-cost nanosatellite development. It is used for evaluating the radiation tolerance of commercial off-the-shelf embedded computers. However, major revisions are required to address these gaps.

  1. Only 15 MeV protons are tested, but LEO orbit proton energy ranges widely. The lack of multi-energy tests makes it impossible to characterize the energy-dependent cross-section of the system.
  2. Only two radiation tests are conducted, with a total of 5 SEUs detected. The small sample size results in large uncertainty in SER calculation, reducing the credibility of the "3 failures/year" conclusion.
  3. The study only focuses on DDR3 SDRAM memory, ignoring other critical components. Why? Other components may also suffer from SEEs, such as eMMC flash, Sitara SoC core, peripheral interfaces like Ethernet/CAN.
  4. The manuscript attributes error bursts to SEUs in SDRAM control registers, but this is only an inference without direct evidence. It cannot rule out other causes.Explain it.
  5. The manuscript proposes spatial redundancy and temporal redundancy but does not experimentally verify their effectiveness.

Author Response

Authors thank you very much for taking the time to review our manuscript and for your valuable comments that helps us to improve our article. Please check the attached file with the response.

Reviewer 2 Report

Comments and Suggestions for Authors

This article examines the effects of proton beams on the operation of a single-board computer. Specifically, it examines errors in RAM caused by proton beam irradiation. A strength of the work is the very detailed description of the object of study, both in terms of hardware and software. The analysis of the detected errors is detailed. The error rate in the real-world use of such systems in low-Earth orbit is also well estimated. However, as the reviewer understood from the article, both tests were performed using the same computer sample. Since chips, including RAM, exhibit some variation in quality during the manufacturing process, the lack of statistics for different samples is, in my opinion, a factor preventing the publication of this work in the journal Sensors.

Minor question:
The sample was exposed to irradiation for approximately 40 minutes and 2 hours. Can the authors specify the time when the errors they recorded occurred?

Author Response

(The authors gave the same response as above.)

Reviewer 3 Report

Comments and Suggestions for Authors

Sensors (ISSN 1424-8220)
Manuscript ID  sensors-3995405
Type  Article
Title  Evaluation by Proton Radiation Tests of a COTS Embedded Computer Running the cFS Flight Mission Software for a Nanosatellite
Authors  Vanessa Vargas *, Pablo Ramos, Alfredo Bautista, Alejandro Castro-Carrera, and Yolanda Morilla

This paper evaluates the radiation tolerance of a commercial off-the-shelf (COTS) embedded computer (BeagleBone Black) operating with NASA’s Core Flight Software (cFS) on the RTEMS real-time operating system, using 15 MeV proton irradiation tests. The study demonstrates the feasibility of using low-cost embedded platforms for nanosatellite on-board computers (OBCs) under low Earth orbit (LEO) radiation environments. The manuscript is technically sound, well-structured, and experimentally thorough. The results are valuable for academic and small-budget space missions. The paper can be accepted after minor revisions, mainly related to clarification, organization, and presentation details.

My detailed comments are as follows:

  1. Novelty clarification:
    The Related Works section demonstrates that the authors are well acquainted with representative studies on radiation effects in COTS devices and have provided a comprehensive literature overview. However, the paper does not clearly articulate how the present work differs from these studies in terms of research angle, methodology, or system-level scope. The Introduction should more explicitly highlight the distinctive aspects of this work compared with previous COTS radiation testing efforts.
  2. Section 2 (Materials and Methods):
    This section is detailed and comprehensive; however, some lengthy hardware and memory address descriptions could be summarized or moved to an Appendix /Supplementary File to improve flow and conciseness.
  3. Discussion section:
    Please include a short paragraph discussing the limitations of the current experiments and potential directions for future work.
  4. Minor formatting issues:

Verify the correct spelling of “architecture” (appears once as “arquitecture” on Page 5).

Author Response

(The authors gave the same response as above.)

Round 2

Reviewer 1 Report

Comments and Suggestions for Authors

It can be accepted in this form.

Reviewer 2 Report

Comments and Suggestions for Authors

The authors improved the text of the article by adding even more details of the experimental study, and also supplemented the article with a discussion of another group's results obtained for DDR5. I still believe the work should be supplemented with experiments on other hardware samples, but I get the feeling the editors of Sensors journal do not consider this necessary for the article's acceptance. For this reason, I believe the article can be accepted for publication.